# Silver Nanoparticles as Modulators of Myogenesis-Related Gene Expression in Chicken Embryos

**DOI:** 10.3390/genes12050629

**Published:** 2021-04-22

**Authors:** Walaa A. Husseiny, Abeer A. I. Hassanin, Adel A. S. El Nabtiti, Karim Khalil, Ahmed Elaswad

**Affiliations:** 1Department of Animal Wealth Development, Faculty of Veterinary Medicine, Suez Canal University, Ismailia 41522, Egypt; drabeer2000@gmail.com (A.A.I.H.); adelelnabtiti@gmail.com (A.A.S.E.N.); ahmed_elaswad@vet.suez.edu.eg (A.E.); 2Anatomy and Embryology Department, Faculty of Veterinary Medicine, Cairo University, Giza 12211, Egypt; karim.khalil@vet.cu.edu.eg

**Keywords:** myogenesis, *MRFs*, embryogenesis, real-time PCR, nanoparticles, broiler

## Abstract

The present study was conducted to investigate the effects of colloidal nanoparticles of silver (Nano-Ag) on the expression of myogenesis-related genes in chicken embryos. The investigated genes included the members of the myogenic regulatory factors family (*MRFs*) and myocyte enhancer factor 2A (*MEF2A*) genes. A total of 200 fertilized broiler eggs (Indian River) were randomly distributed into four groups; non-injected control, injected control with placebo, treatment I in ovo injected with 20 ppm Nano-Ag, and treatment II in ovo injected with 40 ppm Nano-Ag. The eggs were then incubated for 21 days at the optimum temperature and humidity conditions. Breast muscle tissues were collected at the 5th, 8th, and 18th days of the incubation period. The mRNA expression of myogenic determination factor 1 (*MYOD1*), myogenic factor 5 (*MYF5*), myogenic factor 6 (*MYF6*), myogenin (*MYOG*), and *MEF2A* was measured at the three sampling points using real-time quantitative PCR, while MYOD1 protein expression was evaluated on day 18 using western blot. Breast muscle tissues were histologically examined on day 18 to detect the changes at the cellular level. Our results indicate that myogenesis was enhanced with the low concentration (20 ppm) of Nano-Ag due to the higher expression of *MYOD1*, *MYF5*, and *MYF6* at the transcriptional level and *MYOD1* at the translational level. Moreover, histological analysis revealed the presence of hyperplasia (31.4% more muscle fibers) in treatment I (injected with 20 ppm). Our findings indicate that in ovo injection of 20 ppm Nano-Ag enhances the development of muscles in chicken embryos compared with the 40-ppm dosage and provide crucial information for the use of silver nanoparticles in poultry production.

## 1. Introduction

Nanomaterials, nanocarriers, and nanocapsules are examples of nanoproducts used in diverse fields such as the medical [1], veterinary, and agricultural fields. The applications of nanotechnology in livestock and poultry production include optimizing the digestive efficiency, enhancing the immunity, and improving the performance [2]. Gold nanoparticles can boost antioxidant capability, immunity, and efficiency in poultry [3] while chromium nanoparticles increased skeletal muscle mass and improved pork quality [4]. Silver nanoparticles can be used in poultry production as an alternative to the use of antibiotic growth promoter [5].

Nano-Ag have been used in the poultry industry as antimicrobial or nutritional agents [6]. Due to their small size, nanoparticles of silver (Nano-Ag) can penetrate the cells, enter the nucleus, and interact directly with DNA particles or related proteins [7]. Clear evidence indicates that Nano-Ag can modify the gene expression both in vivo and in vitro [8]. For example, Sawosz et al. [9] revealed that Nano-Ag are functional on molecular mechanisms and affect muscle morphology via increasing the number of nuclei per cell and fiber area. Sawosz et al. [10] stated that Nano-Ag are effective on selected genes expression (fibroblast growth factor 2 (*FGF2*), vascular endothelial growth factor A (*VEGFA*), and Na^+^/K^+^ transporting ATPase (*ATP1A1*), and myogenic differentiation 1 (*MYOD1*)) implicated in muscle development and may speed muscle cell growth and maturation. On the contrary, [11] reported that Nano-Ag did not influence the hatchling’s development, but that metabolic rate and fat uptake (FU) were affected.

The myogenic regulatory factors (*MRFs*) are four transcriptional factors which are important for commitment and differentiation of muscle cells in vertebrates [12]. Myogenic factor 5 (*MYF5*), myogenic determination factor 1 (*MYOD1*), myogenin (*MYOG*), and myogenic factor 6 (*MYF6*) are the basic helix-loop-helix family members that monitor skeletal muscle determination and differentiation through embryo development and post-hatch muscle development. *MYF5* controls the entry of the cells into the skeletal muscle program, explaining its early expression [13]. It is expressed before the myogenic fate adoption and is considered a determination factor depending on the information gained from *MYF5* null mice [14]. *MYOD1* is transcribed after *MYF5* starts expression in the hypaxial and then in the epaxial dermomyotome [15]. The expression of *MYOD1* in a *MYF5−/−* (null mutation) promotes cells to the myogenic lineage, although with a lag in certain populations [14]. *MYF6* is expressed in the somitic bud simultaneously, if not before *MYF5*, and is capable of myogenesis induction in *MYF5* and *MYOD1* deficiency, suggesting that *MYF6* has the activity of distinction and determination [14]. *MYOG* is a key gene to motivate the muscle recognition program, unlike *MYOD1* and *MYF5*. *MYOG* is required particularly during the late differentiation of myogenic cells (myoblasts) into myotubes [16]. The *MEF2* family alone does not exhibit myogenic activity but they support the function of *MRFs* through transcriptional co-operation to mediate muscle-specific gene expression [17]. *MRF* and *MEF2* factors actually interact [18] and improve their expression in positive feedback mechanisms [19].

It has been demonstrated that silver nanoparticles could increase the muscle fiber area and the number of nuclei per muscle cell [9]. On a molecular scale, Nano-Ag has been shown to modulate the expression of *FGF2* and *VEGFA* [20]. Muscle development is a complex process that is controlled by several genes such as the *MRFs* family, but the effect of silver nanoparticles on the expression of such genes throughout the embryonic growth phase in chicken have not been comprehensively investigated. Therefore, the objectives of the current study were to assess the modulating effects of Nano-Ag on the expression of *MRFs* and *MEF2A* genes in chicken embryos, and to explore the consequent effects of gene expression alteration on the development of breast muscles.

## 2. Materials and Methods

### 2.1. Nanoparticles

The hydrocolloid solution of Nano-Ag (Nanoworled, Egypt) was produced by the chemical reduction method [21] from silver nitrate salt (AgNO_3_) (99.9%), and the salt solution was dissolved in pure water. The colloidal contained 50 ppm of Nano-Ag, with 9 ± 0.5 nm particle size as evaluated by transmission electron microscope (TEM) [22].

### 2.2. Experimental Design

This experiment was conducted at the Department of Animal Wealth Development, Faculty of Veterinary Medicine, Suez Canal University, Egypt. Fertilized eggs from 44-week-old Indian river chickens (broiler line) were acquired from a commercial hatchery. Upon arrival to the laboratory, the eggs were numbered, weighed (60 ± 2 g), and divided into four groups, namely (1) non-injected control (*n* = 50), (2) injected control (injected with placebo, the diluent used to dissolve Nano-Ag, *n* = 50), (3) treatment I (injected with 20 ppm of Nano-Ag per embryo, *n* = 50), and (4) treatment II (injected with 40 ppm of Nano-Ag per embryo, *n* = 50). The volume of injection was 0.3 mL for the three in ovo injected groups. Directly before incubation, eggs were injected through the hole made by a mechanical driller, and the injection solution was delivered into the air sac between the outer and inner shell membrane at the blunt end using a sterile 1-mL hypodermic syringe with a 28-gauge needle. The site of injection was disinfected with 75% ethanol before and after injection, and the hole was then sealed with adhesive tape. Injected and non-injected eggs from the four groups were artificially incubated for 21 days under optimum conditions of temperature, humidity, and turning. The incubation conditions were 37.8 °C, 65% humidity, and turning once every two hours for the first 18 days, then 37.3 °C and 70% humidity and no turning from day 19 until chicks hatched. This study was performed during the pre-hatch growth period of broiler chicken according to the recommendations of the scientific research ethics committee, Faculty of Veterinary Medicine, Suez Canal University, and was approved with number 2020048.

The embryos were assessed at different stages of development (day 5, 8, and 18 of incubation). The development of embryos was compared with the Hamburger and Hamilton [23] standard. On day 5 and 8 of incubation, sampled eggs (*n* = 72) were broken and the whole embryos were collected and preserved in RNA safeguard (BioFlux, Bioer Technology Co. Ltd., Hangzhou, China) until further analysis. On day 18 of embryogenesis, breast muscles samples (*n* = 36) were dissected and divided into three parts. Parts were preserved in RNA safeguard (BioFlux, Bioer Technology Co. Ltd., Hangzhou, China) for gene expression analysis. For protein analysis, another part of breast muscles (*n* = 36) were collected in labelled cryo-tubes and directly frozen in liquid nitrogen (−196 °C) then preserved at –80 °C. Furthermore, parts of breast muscle tissues (*n* = 36) were collected and preserved in 10% formalin for histological examination.

### 2.3. Gene Expression Analysis Using Real-Time PCR

Whole embryos and breast muscle samples (*n* = 3 samples/group) were homogenized in GeneZol^TM^ CT Reagent (ready-to-use chemical for total RNA extraction from different samples, Puregene, Genetix Biotech Asia Pvt. Ltd., New Delhi, India) using a tissue homogenizer (Hangzhou Miu Instruments Co., Ltd., Zhejiang, China). Total RNA was isolated following the instructions of the manufacturer. RNA samples were quantified by a UV1100 spectrophotometer (TechComp, Hong Kong). Eleven microliters of total RNA were reverse transcribed using reverse transcriptase with oligo (dT)_18_ (Thermo Fisher Scientific, Vilnius Lithuania) and random primers. Real-time PCR was performed with the complementary DNA as a template, gene specific primer (GSP) (designed using the web-based NCBI primer-BLAST tool and synthesized by BIONEER Inc., Alameda, CA USA; Table 1), and Maxima SYBR Green/ROX qPCR Master Mix (Thermo Fisher Scientific, Lithuania) in a StepOnePlus real-time PCR instrument. The reaction mixture consisted of 12.5 µL SYBR Green, 1.5 µL Forward Primer (FP), 1.5 µL Reverse Primer (RP), 4 µL cDNA, and 5.5 µL PCR-grade water. The samples were initially denatured for 15 min at 95 °C and then amplified for 40 cycles of 15 s at 94 °C (denaturation), 60 s at 60 °C (annealing and extension), followed by quantification. A melting curve analysis (95 °C for 15 s, 60 °C for 1 min, and 95 °C for 15 s) was performed to confirm the product specificity. For each cDNA sample, the reaction was conducted in triplicate. β actin gene (*ACTB*) was used as a housekeeping gene for data normalization.

### 2.4. Western Blot Analysis

To evaluate whether Nano-Ag influence on genes involved in myogenesis (day 18 of incubation) at the translational level, protein electrophoretic patterns for transcriptionally upregulated genes were detected and monitored via the SDS-PAGE (sodium dodecyl sulfate polyacrylamide gel electrophoresis, 15%) technique [24]. Soluble proteins were extracted from one embryo (18 days-old) in each group using TriFast (Peqlab, VWR International, LLC, Philadelphia PA, USA) following the manufacturer’s protocol; 50–100 mg of tissue were homogenized in 1 mL TriFast, then the protein was precipitated with 1.5 mL isopropanol at 12,000× *g* for 10 min at 4 °C. The protein pellet was washed three times with 2 mL of 0.3 M guanidinium hydrochloride in 95% ethanol for 20 min at room temperature (RT) then centrifuged at 7500× *g* for 5 min at 4 °C. The protein concentration was determined using T80 spectrophotometer (PG Instruments Limited, Leicestershire, UK) based on previously published protocols [25]. The protein pellet was then dissolved in 1% sodium dodecyl sulfate-polyacrylamide gel. The samples were separated in 15% SDS–PAGE (30 μg protein/lane). Afterward, the samples were blotted onto *Hybond*™ nylon membrane (GE Healthcare, VWR International, LLC, Philadelphia PA, USA) via a TE62 standard transfer tank with a cooling chamber (Hoefer Inc., San Francisco, CA, USA) and incubated for 1 h at RT in the blocking solution. Additionally, β-actin was used as a housekeeping protein. The membrane was incubated overnight at 4 °C in an antibody solution containing anti-MYOD1 (abcam, ab16148, Abcam plc, Cambridge, UK). For data normalization, anti-β-actin primary antibody (abcam, ab228001, Abcam plc, Cambridge, UK) was used. The membrane was washed at RT for 30–60 min then incubated for 1 h at RT in antibody solution with a proper dilution of HRP-conjugated secondary antibody, then washed for 30–60 min. Totallab analysis software, ww.totallab.com, (Ver.1.0.1) was used for data analysis.

### 2.5. Histological Examination of Breast Muscle Tissue

Histological and statistical quantitative analyses of myofibers in breast muscles of the non-injected control group and treated groups were performed (*n* = 3 samples from each group, 3 sections from each sample, 9 total sections for each group). Chicken embryos on day 18 of incubation were decapitated and breast muscle tissues were sampled and preserved in 10% formalin. The samples were dehydrated and paraffin embedded. Serial sections with 6 μm thickness were produced by a rotary microtome. The sections were mounted on microscopical slides and stained with the regressive staining method using Harris hematoxylin (VWR International, LLC, Philadelphia PA, USA) and eosin with phloxine (Sigma-Aldrich, Inc., St. Louis MO, USA). Myofibers were counted and cell numbers were calculated as the fibers number per cross-sectional muscle area with the help of the “Cell Counter” features of ImageJ program. The average size of myofibers was also detected using the same features [26]. All images were obtained using bright field Leica ICC50 HD Microscope with an in-built camera (Leica Microsystems, Wetzlar, Germany).

### 2.6. Statistical Analysis

Data analysis was conducted using a one-way analysis of variance (ANOVA) test generalized linear model method in IMB SPSS statistics version 22.0 to detect the effect of treatment. Duncan’s multiple range test was used to detect the significant difference between groups at *p* < 0.05, and all data were presented as the mean ± standard error (SEM). The target gene expression in the treated groups was normalized to the non-injected control group, normalized to the β actin (the housekeeping gene) [27].
ΔCT (a target sample) = CT (target gene) − CT (reference gene)
ΔCT (a reference sample) = CT (target gene) − CT (reference gene)
ΔΔCT = ΔCT (a target sample) − ΔCT (a reference sample)
2^−ΔΔCT^ = normalized expression ratio

## 3. Results

### 3.1. Gene Expression Analysis

The expression of *MRFs* and *MEF2A* was explored at different stages of development in response to in ovo injection of Nano-Ag on day 1 of incubation (Figure 1). For *MRFs*, the expression of *MYOD1* mRNA on day 5 of incubation was higher in treatment I (20 ppm Nano-Ag) than the non-injected control group but the difference was not significant. On day 8 of incubation, *MYOD1* mRNA expression increased significantly in treatment I and non-significantly in treatment II (40 ppm Nano-Ag) compared with the non-injected control. Similar results were obtained on day 18 of incubation where the expression of *MYOD1* increased in both Nano-Ag treatments compared with the non-injected control, but the difference was significant only in treatment I (Table 2). The expression of *MYF5* mRNA on day 5 of incubation increased non-significantly in treatments I (20 ppm Nano-Ag) and II (40 ppm Nano-Ag) compared with the non-injected control. On day 8 of incubation, *MYF5* mRNA expression increased in both treatments but it was significant only in treatment I. On day 18 of incubation, *MYF5* expression decreased in treatment I with a non-significant increase in treatment II (Table 2).

The expression of *MYF6* mRNA on day 5 and 8 of incubation increased significantly in treatment I (20 ppm Nano-Ag) and treatment II (40 ppm Nano-Ag) while on day 18 of incubation there was no significant difference in either Nano-Ag treated groups (Table 2). The expression of *MYOG* mRNA on days 5, 8, and 18 of incubation showed non-significant differences in treatment I (20 ppm Nano-Ag) and treatment II (40 ppm Nano-Ag) compared with the non-injected control.

The expression of *MEF2A* mRNA decreased non-significantly on day 5 in both treatment I (20 ppm Nano-Ag) and treatment II (40 ppm Nano-Ag). On day 8, the expression also decreased in both treatments but was significant only in treatment II compared with the non-injected control. On day 18, a non-significant increase in *MEF2A* expression was observed in both Nano-Ag treatments (Table 2).

### 3.2. Western Blot Analysis

Comparison of MYOD1 protein expression level with β-actin indicated that the expression levels in both treated groups were higher than in the non-injected control group. The expression levels increased by 1.40 and 1.36 folds in the 20-ppm and 40-ppm Nano-Ag treatments, respectively (Figure 2, Table 3).

### 3.3. Histological Examination

Histological sections from the two treatments and non-injected control group are shown in Figure 3. According to statistical analysis of histological samples from day 18 of incubation (Table 4), the breast muscles of treatment I embryos (20 ppm Nano-Ag) exhibited 31.4% more muscle fibers (4314.11 ± 387.06) than the non-injected control group (3283.22 ± 300.08). Cross-sectional area of muscle fibers did not show significant difference between chicken embryos of the non-injected control group (30,584.33 ± 1978.87 µm^2^) and treated group I (26,722.33 ± 994.7 µm^2^), while there was a significant decrease in treated group II (40 ppm Nano-Ag) (24,214.33 ± 1798.54 µm^2^) than the non-injected control, *p* > 0.05. The average size of muscle fibers was larger in the non-injected control group (10.16 ± 1.31 µm^2^) and treated group II (12.45 ± 1.74 µm^2^) than in treated group I (6.51 ± 0.62 µm^2^), *p* > 0.05. The percentage of muscle fiber area was also higher in the non-injected control group (36.31 ± 2.30%) than treated group I (30.73 ± 1.19%) and treated group II (30.16 ± 1.50%), *p* > 0.05.

### 3.4. Hatchability Results

The effect of in ovo administration of Nano-Ag on hatchability was assessed at the end of the incubation period (after 21 days). The hatched chicks in each group were identified, and the scientific hatchability percent was calculated as shown in Table 5.

## 4. Discussion

In the current study, the eggs were injected with hydrocolloidal solution of Nano-Ag (20 ppm and 40 ppm) and the effects were detected in the treated groups and compared with the non-injected control group. In this experiment, we included a control group injected with a placebo to find whether the mechanical manipulation by injection has any effect compared to the non-injected control group, the perception recommended by [10]. It is the first study that investigated the effect of Nano-Ag on the expression of *MRFs* and *MEF2A* at different embryonic stages (day 5, day 8, and day 18).

The results indicated that there was no effect of mechanical manipulation by injection on the development of embryos during the incubation period in the control group injected with placebo. This result is contrary to other studies where in ovo manipulation, particularly in early embryonic life, was relatively unsuccessful concerning hatchability [28]. Nano-Ag with 20 ppm and 40 ppm concentration in treatment I and II, respectively, up-regulated the expression of *MYOD1* mRNA which is consistent with the finding of [10]. Moreover, Sawosz, et al. [9] reported that Nano-Ag with 50 ppm up-regulated *MYOD1* and *ATP1A1* expression compared with the control group indicating that Nano-Ag activates myogenesis. On the contrary, Sawosz, et al. [29] demonstrated that in ovo administration of 50 ppm Nano-Ag hydrocolloids and composites of Ag with Cu and Pd could not affect mortality, growth, and development of 48 h and 20 days old embryos. This may be due to the nano-size of silver which enables the penetration into tissues or efficient cell uptake, and perfect delivery of active compounds to target cells [30]. Another hypothesis explains how Nano-Ag could improve embryo growth and development by providing the oxygen demand and increasing the metabolism rate [31]. Myogenesis is firmly constrained by the availability of oxygen and nutrient and thus by the metabolic rate, specifically by the oxygen-dependent mechanisms of oxidation and the energy amount stored within the cell [32]. Inadequate amounts of O_2_ in the muscle can restrict its development. Silver can absorb O_2_, and oxygen species were supposed to coexist on its surface [33]. It is anticipated that Nano-Ag have the ability to cross cell membranes and transport O_2_ directly to the muscle cells, functioning as small delivery vehicles [9].

Nano-Ag treatment up-regulated the expression of *MYOD1*, *MYF5*, and *MYF6* throughout the incubation period followed by a down-regulation near the end of incubation, where the peak of expression was achieved on the 8th day of incubation. High levels of *MYF6* expression were detected during the embryonic development in both treatment I and II without significant difference. This result agrees with Saitoh et al. [34] who studied the expression profile of *MYOD1*, *MYOG*, *MYF6,* and *MYF5* in developing chicken breast muscle. In normal conditions, *MYOD1* was detectable as early as day 8 in ovo, reaching a maximum on day 12, then decreased steadily during late embryogenesis. In the case of *MYOG*, it was detected from day 8, and gradually increased up to day 12 in ovo, then greatly reduced after day 12. This means that Nano-Ag increased the gene expression at the same time as the expression profile under normal conditions. *MYOD1* is expressed in proliferating myoblasts and encodes for nuclear proteins, regulating muscle cell differentiation through the stimulation of cell cycle arrest [35]. Nano-Ag increased the expression of *MYOD1* throughout the embryo development at all stages of myogenesis [36]. According to Fong and Tapscott [37], *MYOD1*, *MEF2*, and Six proteins cooperatively interact with each other and with histone-modifying enzymes which control the opening of chromatin to activate target genes.

There are no previous studies on the effect of in ovo injection of Nano-Ag on *MYF5*, *MYF6*, *MYOG*, and *MEF2A* gene expression and their effects on myogenesis during the development of the chicken embryo. The current study revealed that Nano-Ag up-regulated the expression of *MYF5* and *MYF6* and down-regulated the expression of *MYOG* and *MEF2A*. *MRFs* were categorized into early factors (*MYOD1* and *MYF5*) that are involved in the commitment and proliferation of the myogenic directed cells, and late factors (*MYOG* and *MYF6*) which regulate the terminal differentiation of the committed cells [38]. Divided into two categories, these transcription factor roles are redundant to confirm appropriate muscle differentiation in combination with less specific factors as members of the *MEF2* family [39]. A recent study showed that MYF6 can target and reciprocally repress *MEF2,* resulting in decreased muscle growth [40]. Brunetti and Goldfine [41] reported that the transcription of *MYOG* is switched on at the end of embryogenesis, and consequently the process of proliferation decreases while the formation of myofibers becomes a highly active process. In the present study, Nano-Ag down-regulated the expression of *MYOG* in both treatments on days 8 and day 18 of incubation, suggesting that Nano-Ag induces myogenic cell determination through prolonged higher expression of *MYOD1*, *MYF5*, and *MYF6*.

Our protein analysis data by western blot indicated that Nano-Ag with different concentrations (20 ppm and 40 ppm) increased the protein expression of *MYOD1* by 1.40 and 1.36 fold, respectively. This indicates that Nano-Ag regulates myogenesis not only at the transcriptional level but also at the translational level. Several studies such as [9,10,20] detected the effect of in ovo administration of Nano-Ag on the protein expression of *FGF2* and *VEGFA* by using the enzyme-linked immunosorbent assay (ELISA). They demonstrated that a dosage of 50 ppm Nano-Ag increased the expression compared to the control group, while 20 ppm Nano-Ag did not change the protein expression of FGF2 in breast muscles.

In the present study, quantitative measurements of histological samples demonstrated that the number of muscle fibers was higher in treatment I injected with 20 ppm compared with the non-injected control group. Our observation indicated hyperplasia of muscle fibers; but the average size of cells, cross-sectional area, and area percent were lower. On the contrary, the number of cells in treatment II injected with 40 ppm was lower, but the average size of cells was higher than the non-injected control group, indicating the presence of hypertrophy. The observations proved that Nano-Ag stimulated breast muscle growth via hyperplasia (20 ppm dosage) and hypertrophy (40 ppm dosage). These findings agree with the study of Sawosz, et al. [9] where the average cell number per area was lower in all the injected groups compared with the non-injected control group, but the average number of nuclei in proportion to the number of cells was higher indicating a significant role for Nano-Ag in muscle development.

## 5. Conclusions

In ovo administration of Nano-Ag with low concentration (20 ppm) on day 1 of incubation can modulate myogenesis-related gene expression and improve muscle development in chicken embryos without any adverse effects on hatchability. Additional research is needed to follow up the changes in gene expression and muscle development after hatch as these birds grow. This would allow precise assessment of the possibility to use Nano-Ag on a commercial scale to improve poultry production.

## Figures and Tables

**Figure 1 genes-12-00629-f001:**
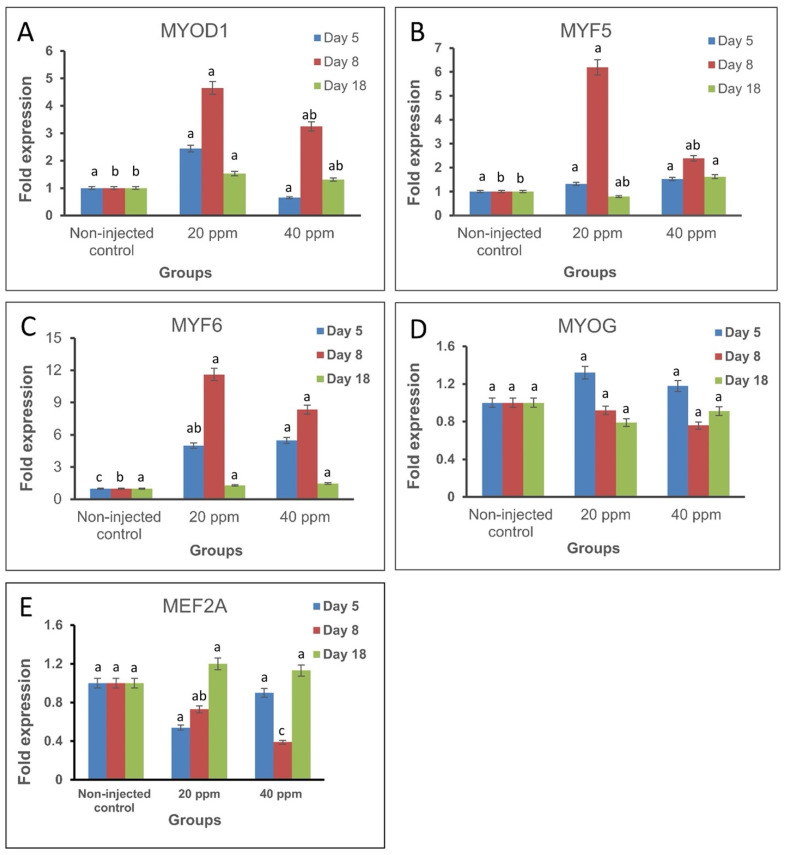
Expression fold change of the studied myogenesis-related genes in breast muscles of chicken embryos (*n* = 3, from each group at each time point) at the three sampling timepoints in the treatment groups (treatment I, 20 ppm Nano-Ag; treatment II, 40 ppm Nano-Ag) compared with the non-injected control group. (**A**): *MYOD1*, myogenic determination factor 1. (**B**): *MYF5*, myogenic factor 5. (**C**): *MYF6*, myogenic factor 6. (**D**): *MYOG*, myogenin. (**E**): *MEF2A*, myocyte enhancer factor 2A. Error bars indicate standard error. The level of significance is *p* < 0.05.

**Figure 2 genes-12-00629-f002:**
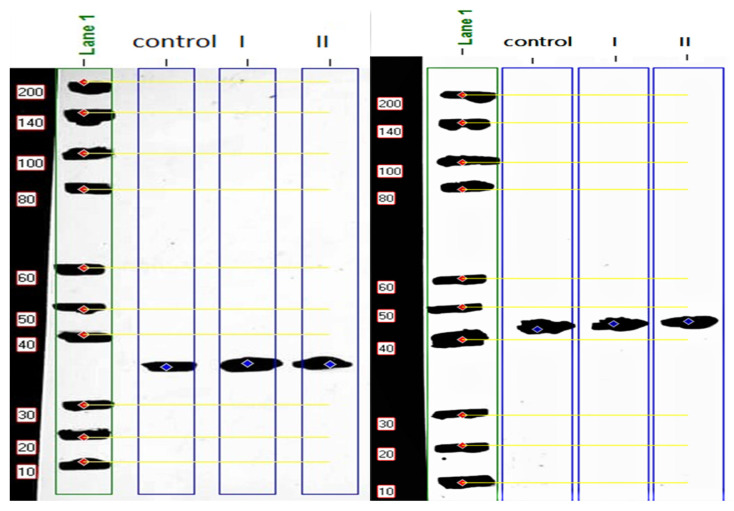
Molecular weight (MW) calculation of myogenic determination factor 1 (MYOD1) (left) and β-actin (right) proteins expression level for the non-injected control samples and treated groups samples using protein ladder (200 kDa). The calculated MW of MYOD1 34.31 kDa.

**Figure 3 genes-12-00629-f003:**
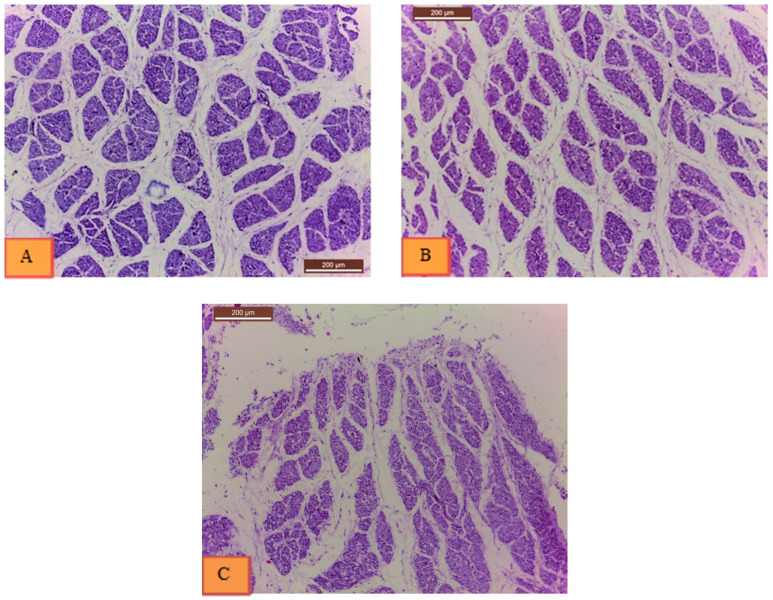
Histological slides from (**A**) non-injected control group, (**B**) treatment I (20 ppm Nano-Ag), and (**C**) treatment II (40 ppm Nano-Ag). Histological examination indicated 31.4% increase in the number of muscle fibers (hyperplasia) in treatment I, and larger average size of muscle fibers in treatment II (hypertrophy) (12.45 ± 1.74 µm^2^).

**Table 1 genes-12-00629-t001:** Gene specific primer sequences.

Gene *	Accession Number	Primer Sequence *	Amplicon Size (bp)	Reference
*MYOD1*	NM_204214.2	FP: 5′ GAATCACCAAATGACCCAAAG 3′ RP: 5′ CTCCACTGTCACTCAGGTTTC 3′	185	This study
*MYF5*	NM_001030363	FP: 5′ AGGAGGCTGAAGAAAGTGAACC 3′ RP: 5′ TAGTTCTCCACCTGTTCCCTCA 3′	155	This study
*MYF6*	NM_001030746	FP: 5′ CCCCTTCAGCTTCAGCCC 3′ RP: 5′ CTCATTTCTCCACCGCCTCTTC 3′	242	This study
*MYOG*	N M_204184	FP: 5′ AATCCTTTCCCACTCCTCTCCA 3′ RP: 5′ TTGGTCGAAGAGCAACTTGG 3′	176	This study
*MEF2A*	NM_204864	FP: 5′ TCGGTGCGAAGTTTTCCTCT 3′ RP: 5′ CTGTTCCGTTCGTCCATTATTC 3′	250	This study
*ACTB*	396526	FP: 5′ GTCCACCTTCCAGCAGATGT 3′ RP: 5′ ATAAAGCCATGCCAATCTCG 3′	169	[20]

***** Abbreviations: myogenic determination factor 1, *MYOD1*; myogenic factor 5, *MYF5*; myogenic factor 6, *MYF6*; myogenin, *MYOG*; myocyte enhancer factor 2A, *MEF2A*; β-actin; *ACTB*. The parameters used for primers design were nucleotide length (18–24 bp), melting temperature (T_m_) (60 °C), GC content (40–60%), and the PCR product length (150–250 bp).

**Table 2 genes-12-00629-t002:** Least square means and their standard errors for the effect of Nano-Ag treatment (20 and 40 ppm) on the expression of myogenic determination factor 1 (*MYOD1*), myogenic factor 5 (*MYF5*), myogenic factor 6 (*MYF6*), myogenin (*MYOG*), and myocyte enhancer factor 2A (*MEF2A*) on days (D) 5, 8, and 18 of incubation. The expression is presented in fold change compared with the non-injected control, (*n* = 3, from each group).

Groups	*MYOD1 **			*MYF5 **			*MYF6 **			*MYOG **			*MEF2A **		
	D 5	D 8	D 18	D 5	D 8	D 18	D 5	D 8	D 18	D 5	D 8	D 18	D 5	D 8	D 18
**Non-injected control**	1.00 ^a^ ± 0.00	1.00 ^b^ ± 0.00	1.00 ^b^ ± 0.00	1.00 ^a^ ± 0.00	1.00 ^b^ ± 0.00	1.00 ^ab^ ± 0.00	1.00 ^c^ ± 0.00	1.00 ^b^ ± 0.00	1.00 ^a^ ± 0.00	1.00 ^a^ ± 0.00	1.00 ^a^ ± 0.00	1.00 ^a^ ± 0.00	1.00 ^a^ ± 0.00	1.00 ^a^ ± 0.00	1.00 ^a^ ± 0.00
**20 ppm**	2.44 ^a^ ± 1.16	4.65 ^a^ ± 0.90	1.53 ^a^ ± 0.18	1.32 ^a^ ± 0.96	6.19 ^a^ ± 2.37	0.79 ^ab^ ± 0.18	5.00 ^ab^ ± 0.47	11.61 ^a^ ± 1.13	1.30 ^a^ ± 0.16	1.32 ^a^ ± 0.18	0.92 ^a^ ± 0.14	0.79 ^a^ ± 0.09	0.54 ^a^ ± 0.48	0.73 ^ab^ ± 0.29	1.20 ^a^ ± 0.22
**40 ppm**	0.65 ^a^ ± 0.08	3.25 ^ab^ ± 1.19	1.31 ^ab^ ± 0.19	1.52 ^a^ ± 0.50	2.39 ^ab^ ± 0.21	1.62 ^a^ ± 0.39	5.49 ^a^ ± 0.70	8.33 ^a^ ± 0.84	1.46 ^a^ ± 0.08	1.18 ^a^ ± 0.31	0.76 ^a^ ± 0.09	0.91 ^a^ ± 0.17	0.90 ^a^ ± 0.14	0.39 ^bc^ ± 0.05	1.13 ^a^ ± 0.38

* Means in the same column with the same superscript letters are not significant (one-way analysis of variance (ANOVA), *p* > 0.05). a and b superscripts indicate significant differences between means.

**Table 3 genes-12-00629-t003:** Data parameters of myogenic determination factor 1 (MYOD1) protein expression level for the non-injected control samples and treated groups samples, (*n* = 1 from each group) normalized to β-actin (reference protein) referred to lane %, and molecular weight (MW) of MYOD1.

Marker			β -Actin			Non-Injected Control			Treatment I			Treatment II		
Band No	Lane %	MW (kDa)	Band No	Lane %	MW (kDa)	Band No	Lane %	MW (kDa)	Band No	Lane %	MW (kDa)	Band No	Lane %	MW (kDa)
1	9.21	200.000	1	98.90	44.47	1	49.98	34.230	1	71.31	34.148	1	68.43	34.314

Lane % is the amount of MYOD1 expression in the target samples divided by the amount of β-actin expression in the same sample.

**Table 4 genes-12-00629-t004:** Least square means and their standard errors for the effect of treatment (Nano-Ag with 20 ppm and 40 ppm) on the count, cross-sectional area (µm^2^), average size (µm^2^), and % area of muscle fibers from breast muscle samples of day 18 of incubation, (*n* = 3 samples from each group, 3 sections from each sample, 9 sections for each group).

Group	Count *	Cross-Sectional Area * (µm^2^)	Average Size* (µm^2^)	% Area *
**Non-injected control**	3283.22 ^b^ ± 300.08	30,584.33 ^a^ ± 1978.87	10.16 ^a^ ± 1.31	36.31 ^a^ ± 2.30
**20 ppm**	4314.11 ^a^ ± 387.06	26,722.33 ^ab^ ± 994.7	6.51 ^b^ ± 0.62	30.73 ^b^ ± 1.19
**40 ppm**	2216.56 ^c^ ± 213.99	24,214.33 ^b^ ± 1798.54	12.45 ^a^ ± 1.74	30.16 ^b^ ± 1.50

* Means in the same column with the same superscript letters are not significant (one-way ANOVA, *p* > 0.05). a and b superscripts indicate significant differences between means.

**Table 5 genes-12-00629-t005:** The number of eggs transported to hatcher, infertile eggs, and unhatched and hatched chicks in each treatment from day 19 until time of hatch.

Groups	Egg Transported to Hatcher (*n*)	Infertile Eggs (*n*)	Unhatched Chicks (*n*)	Hatched Chicks (*n*)	Scientific Hatchability %
**Injected control**	25	3	0	22	22/25 = 88.0
**Treatment I (20 ppm)**	24	3	1	20	20/21 = 95.23
**Treatment II (40 ppm)**	27	0	3	24	24/27 = 88.88

## Data Availability

Data generated in the current study are available from the corresponding author upon request.

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
