# Peer review of "Silver Nanoparticles as Modulators of Myogenesis-Related Gene Expression in Chicken Embryos"

_genes, 2021, doi:10.3390/genes12050629_

Round 1

Reviewer 1 Report

Genes-1136593: Silver nanoparticles as modulators of myogenesis-related gene 2 expression in chicken embryos

 Abstract:  The last statement appears vague.

Introduction:  Much of the introduction is out of context, unfocused; hence, can be reduced significantly.

Provide a statement rationale or hypothesis as to why the authors consider that exposing the embryo to silver nano particle  would modulate muscle growth,  immunity or other biological features?  If this was an exploratory study, there needs to be basis of it. 

Lines 48-58:  This aspect of introduction can be shortened since it has little relevance.  You may start with that “Since muscle development is regulated by….genes, the objective was to find how silver nano particle affects…” or similar statement.  The detailed mechanisms of chicken muscle development are not necessary.

Results:  Tables:  What are the sample numbers (n=?) for each statistical point.  Provide the information in methods and in the legends of each Table.

Table 6:  How many sections of muscle from how many eggs for each treatment group were evaluated for this Table?

Line 298:  Replace: ‘detect’ with “to find whether…”

Line 312: Delete “when” and replace “compared with”

Lines 302-303:  what do you mean by “improving broiler production”.  Which way? Hatchability or growth parameters?

Figure 2 and Table 5:  Provide more information about electrophoresed protein samples whether they were derived from 5, 8, or 18 days-old embryos and the statistical results are obtained from how many scans?  You may also provide this information in the “methods” text.

Conclusion:  There is no hatchability data given anywhere; hence this statement is inaccurate; in all cases the eggs appear to have been broken.  Besides to conclude that the nanoparticle treatment may improve poultry production (eg: muscle growth), did you obtain the weights of the embryos with and without treatment?  Provide data to qualify these statements.

Author Response

Thank you for reviewing our research. We appreciate your time, effort, and constructive comments. We have addressed all your comments and suggestions in the revised version of the manuscript as shown below.

Point 1: The last statement appears vague (Our findings indicate that in ovo injection of 20 ppm Nano-Ag may improve the growth of chicken embryos compared with the 40-ppm dosage and provide crucial information for the applications of silver nanoparticles in poultry production).

Response 1: The last statement means that the minimum effective dose of silver nanoparticles affecting myogenesis is 20 ppm. This is in comparison with the higher dose 40 ppm. We have edited this statement to be more specific (lines 28-30)

Point 2: Introduction:  Much of the introduction is out of context, unfocused; hence, can be reduced significantly.

Response 2: Introduction has been focused and the out-of-context statements have been deleted.

Point 3: Provide a statement rationale or hypothesis as to why the authors consider that exposing the embryo to silver nano particle would modulate muscle growth, immunity, or other biological features?  If this was an exploratory study, there needs to be basis of it. 

Response 3: Thank you for your comment. We have edited this paragraph to provide the rationale for the study (lines 94-102).

Point 4: Lines 48-58:  This aspect of introduction can be shortened since it has little relevance.  You may start with that “Since muscle development is regulated by….genes, the objective was to find how silver nano particle affects…” or similar statement.  The detailed mechanisms of chicken muscle development are not necessary.

Response 4: Lines 56-66 have been deleted and started with “The myogenic regulatory factors (MRFs) are four…”. Also, the detailed mechanisms of chicken muscle development had been removed.

Point 5: Results:  Tables:  What are the sample numbers (n=?) for each statistical point.  Provide the information in methods and in the legends of each Table.

Response 5: The sample numbers had been added in each statistical point in methods and in the legends of each table.

Point 6: Table 6:  How many sections of muscle from how many eggs for each treatment group were evaluated for this Table?

Response 6: Three sections of muscle from 3 eggs for each treatment group were evaluated. We added this information to the manuscript (line 347).

Point 7: Line 298:  Replace: ‘detect’ with “to find whether…”

Response 7: “Detect” had been replaced with “find whether”, line 367.

Point 8: Line 312: Delete “when” and replace “compared with”

Response 8: “when” had been deleted and “compared to” replaced with “compare with”, line 381.

Point 9: Lines 302-303:  what do you mean by “improving broiler production”.  Which way? Hatchability or growth parameters?

Response 9: Thank you for your comment. This sentence has been deleted to avoid confusion, lines 371-372.

Point 10: Figure 2 and Table 5:  Provide more information about electrophoresed protein samples whether they were derived from 5, 8, or 18 days-old embryos and the statistical results are obtained from how many scans?  You may also provide this information in the “methods” text.

Response 10: The electrophoresed protein samples were derived from 18 days-old embryos as mentioned in experimental design of materials and methods section, lines 183-184.

Point 11: Conclusion:  There is no hatchability data given anywhere; hence this statement is inaccurate; in all cases the eggs appear to have been broken.  Besides to conclude that the nanoparticle treatment may improve poultry production (eg: muscle growth), did you obtain the weights of the embryos with and without treatment?  Provide data to qualify these statements.

Response 11: Thank you for your comment. We have edited this section so that the conclusions are supported by our results. Also, the hatchability data were added in the results section, lines 355-358.

Reviewer 2 Report

Introduction:

Line 33-34: Growing interest in application of nanotechnology in which fields? Can the Authors specify?

37-38: Please, explain how the nanomaterials can optimize the livestock performance

Line 45: Please elaborate, how Nano-Ag is reported to influence muscle development and gene expression? Name genes that have been proven to be affected by Nano-Ag (see comments below)

86-87: Since there are no prior indications that may link the Nano-Ag with MRF genes expression, how was the hypothesis formulated (there is actually no hypothesis – please add one)? Please specify the literature linking Nano-Ag with myogenesis, so that the research goals and hypothesis could be justified. At the moment, I am not convinced that study should be done simply because no one ever done it. Please, elaborate.

Materials and Methods

Line 132: Add brand of tissue homogenizer

Line 134: Add brand of UV1100 spectrophotometer

Line 137: reference error

Table 1: What is the source of primer sequences? If they were designed in-house, please add the method (software) used for design and what parameters were used for design? What was efficiency of the gene expression?

Line 187: Did the Authors use any post-hoc tests?

Line 193: It is obviously not the full formula to calculate fold induction. Please, complete

Lines 194-196: Move ethical statements to beginning of the M&M where animal treatment is described

Results:

Are the raw data (including melting curves) available as supplementary files? If not, please make them available

Figure 1: Figure one doesn’t seem to be in journal quality (the letters are blurred), is it really 300dpi? I see no statistics on the graphs (please mark the differences)

Table 2, Table 3, and Table 4: should be redesigned so that they are merged into one table (they contain the same data for different genes). There is no information in the table what is the variation measure

Line 273: reference error

Figure 3: Picture C is quite different (lower quality) from A and B

Discussion:

Lines 299-300 (and across the text): correct the in-text reference

Lines 302-303: How was this study a follow up to improved broiler performance? To my understanding, the Authors did not demonstrate it yet, if the differences on the embryonic level will have implications on the performance of the broilers post-hatching

Line 307: Where are hatchability results??? Please report them!

Conclusions:

Re-write conclusions completely so they reflect the data presented in this paper

Lines 389-390: The conclusion on hatchability is not reflected by any of the data presented

Line 392: We don’t know if this method increase meatiness. Please conclude only what is supported by the data!!!!

Line 393: I think that at the moment this technique is very far from commercialization and for sure it was not the topic of this paper

Author Response

Thank you for reviewing our research. We appreciate your time, effort, and constructive comments. We have addressed all your comments and suggestions in the revised version of the manuscript as shown below.

Point 1: Line 33-34: Growing interest in application of nanotechnology in which fields? Can the Authors specify?

ʉ۬

Response 1: Thank you for your comment. We have edited this sentence for clarity (lines 34-37).

Point 2: 37-38: Please, explain how the nanomaterials can optimize the livestock performance.

Response 2: Different nanoparticles have been studied for their effects as supplementation in poultry production; nanoselenium, nanogold, nanosilver, chromium nanoparticles. Nano-Se was effective in improving the productive performance and GSH-Px activity of layer and producing Se enriched egg. Nanogold could improve the antioxidant capacity, immunity and performance in poultry. Silver nanoparticles as antimicrobial agents may be used in poultry production as an alternative to the use of antibiotic growth promoter. Chromium nanoparticles increased skeletal muscle mass and improved pork quality. Lines 40-44.

Point 3: Line 45: Please elaborate, how Nano-Ag is reported to influence muscle development and gene expression? Name genes that have been proven to be affected by Nano-Ag (see comments below). 

Response 3: How Nano-Ag influence muscle development is explained in discussion, line 383-386. The genes proven to be affected by Nano-Ag are mentioned in line 49-52.

Point 4: 86-87: Since there are no prior indications that may link the Nano-Ag with MRF genes expression, how was the hypothesis formulated (there is actually no hypothesis – please add one)? Please specify the literature linking Nano-Ag with myogenesis, so that the research goals and hypothesis could be justified. At the moment, I am not convinced that study should be done simply because no one ever done it. Please, elaborate.

Response 4: A good comment, thank you. We have modified this part to clarify the rationale of the experiment (lines 94-101).

Point 5: Line 132: Add brand of tissue homogenizer.

Response 5: The brand of tissue homogenizer is (Hangzhou Miu Instruments Co., Ltd., Zhejiang, China), line 150.

Point 6: Line 134: Add brand of UV1100 spectrophotometer.

Response 6: The brand of UV 1100 spectrophotometer is (TechComp, Hong Kong), line152.

Point 7: Line 137: reference error.

Response 7: Yes, you are right. We have corrected it, line 157.

Point 8: What is the source of primer sequences? If they were designed in-house, please add the method (software) used for design and what parameters were used for design? What was efficiency of the gene expression?

Response 8: The primers were designed using the web-based NCBI primer-BLAST tool, line 156. The parameters used for design were nucleotide length (bp), melting temperature (Tm), GC content (%) and the PCR product length, lines 176-177. The efficiencies near 100% and within 5% of target and reference genes.

Point 9: Did the Authors use any post-hoc tests? 

Response 9: Yes, we used Duncan's multiple range test, line 219-220.

Point 10: Line 193: It is obviously not the full formula to calculate fold induction. Please, complete

Response 10: Thank you. The full formula to calculate the change in fold expression had been added, line 224-227.

Point 11: Lines 194-196: Move ethical statements to beginning of the M&M where animal treatment is described.

Response 11: The ethical statements had been moved to the beginning of M&M, line 129-132.

Point 12: Are the raw data (including melting curves) available as supplementary files? If not, please make them available.

Response 12: We have updated the data availability statement so that all the raw data are available from the corresponding author upon request (lines 484-485).

Point 13: Figure 1: Figure one doesn’t seem to be in journal quality (the letters are blurred), is it really 300dpi? I see no statistics on the graphs (please mark the differences).

Response 13: Yes, it is 300 dpi, and we will provide 600 dpi in the final version. It is very clear. We have added the superscript letters to the figure to show the differences.  

Point 14: Table 2, Table 3, and Table 4: should be redesigned so that they are merged into one table (they contain the same data for different genes). There is no information in the table what is the variation measure.

Response 14: The tables 2, 3, and 4 had been merged in one table “Table 2”, line 280. It provides information about the change in fold expression of MRFs genes throughout the incubation period with superscript from Duncan’s multiple range post-hoc test.

Point 15: Line 273: reference error.

Response 15: Thank you, we have corrected it, line 326.

Point 16: Figure 3: Picture C is quite different (lower quality) from A and B.

Response 16: The quality of Picture C had been improved.

Point 17: Lines 299-300 (and across the text): correct the in-text reference.

Response 17: The in-text references had been corrected, lines 369, 380, 434, 435.

Point 18: Lines 302-303: How was this study a follow up to improved broiler performance? To my understanding, the Authors did not demonstrate it yet, if the differences on the embryonic level will have implications on the performance of the broilers post-hatching.

Response 18: This sentence has been deleted to avoid confusion, lines 371-372.

Point 19: Line 307: Where are hatchability results??? Please report them!

Response 19: The hatchability results have been added, line 355.

Point 20: Re-write conclusions completely so they reflect the data presented in this paper.

Response 20: Thank you. We have rewritten the conclusions so that they are supported by the results.

Point 21: Lines 389-390: The conclusion on hatchability is not reflected by any of the data presented.

Response 21: The data of hatchability had been added in the results section, line 355. We have also rewritten the conclusions so that they are supported by the results. 

Point 22: Line 392: We don’t know if this method increase meatiness. Please conclude only what is supported by the data!!!!

Response 22: This sentence was deleted.

Point 23: Line 393: I think that at the moment this technique is very far from commercialization and for sure it was not the topic of this paper.

Response 23: We have rewritten the conclusions and deleted this statement.

Round 2

Reviewer 1 Report

Lines 34-36 can be removed and the reference incorporated in the 2nd.

Line 42: Delete “in addition”.  The gold nano particles can be used…

 Line 95: Delete ”number” after muscle cell.

Line 98: On molecular scale nano-Ag has been shown to modulate….

Line 134: …different stages of development…..

Line 147: n=3 samples/group

Line 181: to evaluate whether Nano-Ag influences…

Lines 184-185:  Rewrite such as “soluble proteins were extracted from one embryo (?) in each group using…

Question: Based on the sample size this measurement may be inconsequential.  Why did you choose only d18 embryos for this study while d5 or d8 embryos which showed more significant changes in MyoD gene expression perhaps, could have given you better protein results?

Lines 190-192: Provide information as to how the extracted protein was measured before they were subjected to electrophoresis.

Lines 254, 256, and 257: Reword and rewrite since numerical increase is not significant, there is no change.  

Table 3 and Fig 2: Which band is b-actin?

Lines 400-401: Since your gene expression comparisons were only within one stage group under 2 different treatments (vertical), it may not be appropriate to discuss whether the changes in gene expressions, observed between age groups within a treatment are relevant because you did not compare such gene expressions were similar in the control groups at different ages.

Line 450: “agree”

Author Response

Response to Reviewer 1 Comments
Thank you for reviewing our research. We appreciate your time, effort, and constructive comments. We have addressed all your comments and suggestions in the revised version of the manuscript as shown below.
Point 1: Lines 34-36 can be removed and the reference incorporated in the 2nd.
Response 1: The lines had been removed and the reference has been incorporated in the 2nd, line 37.
Point 2: Line 42: Delete “in addition”. The gold nano particles can be used…
Response 2: “in addition” had been deleted, line 41.
Point 3: Line 95: Delete “number” after muscle cell.
Response 3: “number” had been deleted, line 73.
Point 4: Line 98: On molecular scale nano-Ag has been shown to modulate….
Response 4: This sentence had been modified, Line 73.
Point 5: Line 134: …different stages of development…..
Response 5: “embryonic” had been deleted and become different stages of development, line 107. Point 6: Line 147: n=3 samples/group
Response 6: This description had been added, line 120.
Point 7: Line 181: to evaluate whether Nano-Ag influences…
Response 7: The part had been modified, line 154.
Point 8: Lines 184-185: Rewrite such as “soluble proteins were extracted from one embryo (?) in each group using…
Response 8: This part had been modified to start with “soluble proteins were extracted from one embryo”, lines 157-158.
Point 9: Question: Based on the sample size this measurement may be inconsequential. Why did you choose only d18 embryos for this study while d5 or d8 embryos which showed more significant changes in MyoD gene expression perhaps, could have given you better protein results?
Response 9: We depend on the previous studies that perform protein expression detection at the end of incubation period as Filip Sawosz (2012) study “Nano-nutrition of chicken embryos. The effect of silver nanoparticles and glutamine on molecular responses, and the morphology of pectoral muscle”. This was more relevant to our study especially we intended to demonstrate the effect of Nano-Ag on breast muscles histology. In addition, on days 5, and 8 of incubation, the embryo size was small enough to become impossible to obtain an adequate amount of breast muscles for protein analysis.
Point 10: Lines 190-192: Provide information as to how the extracted protein was measured before they were subjected to electrophoresis.
Response 10: Required information about protein fraction quantification had been added, lines 163-165.
Point 11: Lines 254, 256, and 257: Reword and rewrite since numerical increase is not significant, there is no change.
Response 11: This part had been modified to indicate that numerical increase is not significant, Lines 221-226.
Point 12: Table 3 and Fig 2: Which band is b-actin?
Response 12: Figure of B-actin bands in the three experimental groups had been added to figure 2, line 256. Also, band number, lane %, and molecular weight of B-actin in the groups had been inserted in table 3, line 264.
Point 13: Lines 400-401: Since your gene expression comparisons were only within one stage group under 2 different treatments (vertical), it may not be appropriate to discuss whether the changes in gene expressions, observed between age groups within a treatment are relevant because you did not compare such gene expressions were similar in the control groups at different ages.
Response 13: Thank you for your comment, this sentence had been deleted, lines 333 and 334.
Point 14: Line 450: “agree”
Response 14: “Agreed” had been corrected to agree, line 383
